# In-Silico Drug Toxicity and Interaction Prediction for Plant Complexes Based on Virtual Screening and Text Mining

**DOI:** 10.3390/ijms231710056

**Published:** 2022-09-02

**Authors:** Feng Zhang, Kumar Ganesan, Yan Li, Jianping Chen

**Affiliations:** 1School of Chinese Medicine, Li Ka Shing Faculty of Medicine, The University of Hong Kong, 10 Sassoon Road, Pokfulam, Hong Kong, China; 2Shenzhen Institute of Research and Innovation, The University of Hong Kong, Shenzhen 518000, China

**Keywords:** herbal bioinformatics, in-silico toxicity prediction, drug-drug interaction, ligand-based virtual screening, synergism, triple-negative breast cancer

## Abstract

Potential drug toxicities and drug interactions of redundant compounds of plant complexes may cause unexpected clinical responses or even severe adverse events. On the other hand, super-additivity of drug interactions between natural products and synthetic drugs may be utilized to gain better performance in disease management. Although without enough datasets for prediction model training, based on the SwissSimilarity and PubChem platforms, for the first time, a feasible workflow of prediction of both toxicity and drug interaction of plant complexes was built in this study. The optimal similarity score threshold for toxicity prediction of this system is 0.6171, based on an analysis of 20 different herbal medicines. From the PubChem database, 31 different sections of toxicity information such as “Acute Effects”, “NIOSH Toxicity Data”, “Interactions”, “Hepatotoxicity”, “Carcinogenicity”, “Symptoms”, and “Human Toxicity Values” sections have been retrieved, with dozens of active compounds predicted to exert potential toxicities. In *Spatholobus suberectus* Dunn (SSD), there are 9 out of 24 active compounds predicted to play synergistic effects on cancer management with various drugs or factors. The synergism between SSD, luteolin and docetaxel in the management of triple-negative breast cancer was proved by the combination index assay, synergy score detection assay, and xenograft model.

## 1. Introduction

For the treatment of some advanced cancer such as triple-negative breast cancer (TNBC), there are still seldom medications, but chemotherapeutic drugs can achieve moderate effects on patient overall survival according to the latest therapy guidelines and clinical trials [1,2,3]. More than 20 potential severe adverse events can be incurred during chemotherapy [4]. So, it’s important to discover strategies to reduce the negative effects of chemotherapy to improve patients’ quality of life. The synergistic effects of natural substances combined with chemotherapy medications may shed light on this [5,6,7]. Moreover, the paradigm shift from a “one-target, one-drug” mode to a “network-target, multiple-component-therapeutics” mode like network pharmacology will offer much more potential for cancer management. Nonetheless, conventional network pharmacology is based on simple additivity of the potencies and efficacies of individual active compounds of herbal medicines [8,9]. There are potentially super-additive (synergistic) effects and subadditivity (antagonism) in terms of drug combination [10], which may cause unexpected pharmacologic or clinical responses. Plus, conventional network pharmacology is lacking in toxicity analysis for plant complexes. Pharmacological and toxicological assessments for compound combinations by experiments would be exceedingly time- and cost-intensive. Similar compounds may be conferred by similar bioactivities [11]. This fundamental idea allows for the integration of chemical informatics and bioinformatics tools into hypotheses constructs for drug discovery. Advancements in Big Data management have opened up access to massive public datasets and powerful tools, e.g., virtual screening. In-silico drug targets, toxicity, and drug-drug or drug-food interaction predictions based on chemical similarity through virtual screening and machine learning will offer us many preliminary outcomes efficiently for natural products research.

The concept and framework of network toxicity in Traditional Chinese Medicine (TCM) were first proposed by Fan et al. in 2011. They claimed to employ network pharmacology approaches to reconstruct the network of “compound-protein/gene-toxicity” to identify dangerous chemicals and anticipate the harmful side effects of existing compounds [12]. However, no successful practice has been reported based on this concept, which may because of the knowledge gap between compound-target and compound toxicity. And limited information about the toxicity and drug interactions of natural compounds can be found in the public databases for model training. But there are many expert systems (DEREK [13], AMBIT [14], DSSTox [15], Derek Nexus [16], Meteor [17], HazardExpert [18], PASS [19], cat-SAR [20], Toxmatch [21], VEGA [22], ChemIDplus [23]) built for toxicity prediction of synthetic drugs [24]. All of them are based on two different methodologies: The quantitative structure-activity relationship (QSAR) and molecular docking [25]. There are huge knowledge gaps between the molecular docking result and drug toxicity for the distinguished roles of one protein in systematic toxicity and the roles of compound-protein interactions in the activation or degradation of proteins are elusive. So, QSAR-based systems are the most employed computational approaches to predict drug toxicity. However, most QSAR-based systems employing models trained by datasets of FDA-approved synthetic drugs may show less confidence in the prediction of natural products. The performance of some expert systems for toxicity prediction such as TOPKAT, DEREK, and HazardExpert has been reported to be poor [26,27,28,29]. In addition, most toxicity prediction systems only offer limited endpoint alerts without any insight interpretation. Hence, this study aims to construct a prediction workflow (Figure 1) for natural products to alert comprehensive endpoints with detailed information for the first time. Although there are no qualified datasets for “one-step” alert model training for toxicity and drug interaction prediction of natural products. Many models for quantitative structure-activity relationship analysis are available. Firstly, based on one of the best QSAR models and platforms, the basic information about the similar bioactivity compounds of active compounds can be mined. Then the comprehensive toxicity and drug interaction information of similar compounds was collected as the indicator for active compounds. The toxicity and drug interaction prediction can be conducted based on a reasonable similarity score threshold.

PubChem (https://pubchem.ncbi.nlm.nih.gov/) (accessed on 31 December 2021) is the world’s largest repository of publicly available chemical data at the National Institutes of Health. It provides detailed information on chemical and physical properties, biological activity, safety and toxicity, patents, and literature citations. It houses data on about 111 million chemicals, 295 million bioactivities, 34 million publications, and 42 million patents [30]. So, it is the most popular database for chemical-related data mining with detailed research protocols and insights. SwissSimilarity (http://www.swisssimilarity.ch/) (accessed on 30 December 2021) is run by the Molecular Modelling Group of the SIB Swiss Institute of Bioinformatics and the University of Lausanne. It is a user-friendly tool for ligand-based virtual screening from several libraries of small compounds using various methodologies. This platform can work based on various structure descriptors such as FP2 [31], ECFP4 [32], MHFP6 [33], Pharmacophore [34,35], ErG [36], Electroshape [37], and E3FP [38,39]. Even an in-house combined model on this platform can be available to show better performance for structure-activity prediction compared to an independent fingerprint similarity-based or shape similarity-based prediction model [40].

## 2. Results

### 2.1. Similarity Score Threshold Analysis

To build a workable and dependable in-silico prediction workflow, the similarity threshold setting plays an important role in the balance of prediction precision and prediction yield. Presumably, there is a positive correlation between the similarity score and a true prediction, which means a “strict” threshold will eliminate the false predictions. But it does not mean that a “strict” similarity threshold is better than a “low” similarity threshold in terms of prediction yield because there is a significantly negative correlation between the number of active compounds predicted and the similarity threshold (Figure 2). The similarity score in this workflow was regarded as a kind of descriptor of active compounds but not the final probability of a prediction model. Further analysis will be made based on the similarity threshold.

Twenty herbal medicines, 495 active compounds, and 84,056 similar compounds were involved in similarity threshold analysis. The lower the similarity threshold is the more distinct retrieves can be collected. There is a significantly negative correlation between similar compounds retrieved and the similarity threshold (Figure 2a). There is a significantly negative correlation between the number of active compounds with similar compounds retrieved and the similarity threshold when the similarity score is higher than 0.56 (Figure 2b). Similarly, there is a linear (negative) correlation between the total toxicity information retrieved and the similarity threshold (Figure 3a). And there is a significantly negative correlation between the number of active compounds with toxicity prediction retrieved and the similarity threshold.

However, elusive information or even contrast information about the toxicities of the same compound can be retrieved due to a low similarity score. For example, in Table 1, the first elusive retrieve regarding hepatotoxicity of 3-Hydroxystigmast-5-en-7-one compared to the retrieve with the largest similarity score occurred when the similarity score decreased to 0.498, while the first contrast retrieves for beta-sitosterol and campesterol occurred when the similarity score decreased to 0.588 and 0.594, respectively. Herein, we introduced two different concepts: FEP-SS and FCP-SS (Defined in Box 1) which can be utilized to set an optimal similarity score threshold for predictions to balance the prediction precision and yield. The FCP-SS of one active compound is the threshold for retrieves without any controversial results, while the FEP-SS is the threshold for that all the retrieves are consistent. Based on two different toxicity aspects: hepatotoxicity and carcinogenicity, the optimal similarity score threshold analysis for consistent retrieves was conducted. The representative FEP-SS and FCP-SS data of four different herbal medicines can be referred to in Table 2. The sum of all the FEP-SS and FCP-SS values of 20 herbal medicines can be referred to in Appendix A. Low FCP-SS and FEP-SS values probably were gained because of insufficient information from PubChem, herein, values of FCP-SS or FEP-SS ≤ 0.3 were excluded in the descriptive analysis of FCP-SS and FEP-SS values (Figure 4a). Generally, the FCP-SS is numerically less than FEP-SS (Figure 4b). The mean value of FCP-SSs and FEP-SSs are 0.6171 and 0.6181, respectively, and the third quartile value of FCP-SS is 0.759. Given the prediction yield and precision, the similarity score threshold of this prediction project was set at the concentrated value of FCP-SS, the representative value of FCP-SS, 0.6171.

### 2.2. Toxicity Prediction Interpretation

After the construction of the prediction workflow and similarity threshold setting, the toxicity and drug interaction predictions were made for several herbal medicines. From the PubChem database, for the four representative herbal medicines, thirty-one different sections of toxicity information of active compounds such as “Acute Effects”, “NIOSH Toxicity Data”, “Interactions”, “Hepatotoxicity”, “Evidence for Carcinogenicity”, “Symptoms”, “Human Toxicity Values”, and “TSCA Test Submissions” sections have been retrieved (See Appendix A, with 26 active compounds predicted to exert various potential toxicities.

(20S)-Dammar-24-ene-3beta,20-diol 3-acetate was predicted to possess potential hepatotoxicity and reproductive developmental toxicity. Cajinin, calycosin, formononetin, glyzaglabrin, hederagenin, jaranol, kaempferol, luteolin, odoratin, olitoriside, psi-Baptigenin, quercetin, 3-Hydroxystigmast-5-en-7-one and so on may have potentially reproductive developmental toxicity. Olitoriside is an analog of digoxin with a relatively high similarity score of 0.992, which means it may also exert similar toxicity to digoxin. In addition, 3-Hydroxystigmast-5-en-7-one, glycyrrhiza flavonol A, isolicoflavonol, jaranol, kaempferol, luteolin, isoflavanone, and quercetin are probably genotoxic predicted by the toxicity information retrieve of their similar compounds, respectively. Presumably, reproductive and developmental toxicity should be paid enough attention to the consumption of all these four herbal medicines. It is also warranted that digoxin toxicity may occur when *Fructus ligustri* Lucidi is taken at toxic doses. Besides, the prediction results of acute toxicity, antidote and emergency treatment, protein binding, ecotoxicity values, ongoing test status, skin symptoms, eye symptoms, and target organs of some active compounds can be available in Appendix A.

### 2.3. Drug Interaction Prediction Interpretation

The prediction of drug interaction of plant complexes is another important role of this workflow. There are 41 different active compounds involved in the 4 representative herbal medicines, of which the detailed information can be referred to Appendix A. Many active compounds may have significant super-additive or sub-additive effects on drug pharmacokinetics, cancer management, cell survival, drug-induced reproductive developmental toxicity, antibacterial, anticoagulation, or/and cardiovascular function.

For cancer management, there are roughly seven different activities (Enhanced radiotherapy, metastasis inhibition, carcinogenesis inhibition, enhanced chemotherapy, enhanced genotoxicity, enhanced bioavailability, and weakened target therapy) influenced by potential drug interactions predicted based on the similarity score threshold of 0.6171. Cajinin, calycosin, formononetin, glyzaglabrin, isotrifoliol, luteolin, odoratin, and psi-Baptigenin may potentiate the sensitivity of cancer cells to ionizing radiation. Isorhamnetin, isotrifoliol, jaranol, kaempferol, luteolin, quercetin, and sitosterol were predicted with the same similar compound, apigenin or lupeol, to inhibit cancer metastasis. Hederagenin may inhibit carcinogenesis by 1,2-dimethyl-hydrazine, 12-O-tetradecanoylphorbol 13-acetate, or azoxymethane. Gadelaidic acid and icos-5-enoic acid were predicted to inhibit carcinogenesis caused by methyl nitrosourea. Glycyrrhiza flavonol A, 8-C-alpha-L-arabinosylluteolin, isolicoflavonol, isorhamnetin, jaranol, kaempferol, luteolin, and quercetin probably suppress UV-induced skin tumorigenesis. 8-C-alpha-L-arabinosylluteolin, Glycyrrhiza flavonol A, hederagenin, isorhamnetin, isotrifoliol, jaranol, licochalcone B, kaempferol, liquiritin, luteolin, mairin, olitoriside, quercetin, and sitosterol may have potential synergistic effects when treated in combination with many chemotherapeutic drugs on cancer management.

However, mairin may also be a promoter of N-Nitrobis(2-hydroxypropyl)amine and N-methyl-N′-nitro-nitrosoguanidine triggered cancer progression. There is a theoretical risk of enhanced genotoxicity using cisplatin with isorhamnetin, isotrifoliol, kaempferol, luteolin, quercetin, or 8-C-alpha-L-arabinosylluteoli. In addition, weakened target therapy of bortezomib may occur due to the combined treatment of 8-C-alpha-L-arabinosylluteolin, glycyrrhiza flavonol A, isorhamnetin, jaranol, kaempferol, luteolin, or quercetin.

### 2.4. Synergism Detection

Based on the drug interaction prediction, in *Spatholobus suberectus* Dunn (See Figure 5), 3-Hydroxystigmast-5-en-7-one, 8-C-alpha-L-arabinosylluteolin, beta-sitosterol, cajinin, calycosin, campesterol, formononetin, luteolin, and psi-Baptigenin (9 out of 24 screened active compounds) may play synergistic effects on cancer metastasis inhibition, carcinogenesis inhibition, chemotherapy, or/and radiotherapy with various chemotherapeutic drugs or factors. Cajinin, calycosin, luteolin, and psi-Baptigenin, similar to kaempferol, were predicted to enhance the chemotherapeutic drug bioavailability because of the inhibition effects on P-glycoprotein (P-gp) and cytochrome P450 (CYP) (See Appendix A). These two enzyme families play important role in the neutralization and effluxion of various chemotherapeutic drugs including docetaxel [41,42,43], the first-line chemotherapeutic drug in TNBC management [44]. Collectively, there may be potential synergism between SSD and docetaxel in TNBC therapy. Luteolin, extremely similar to apigenin, myricetin, genistein, and kaempferol which have been proved to play synergistic effects on cancer metastasis inhibition, carcinogenesis inhibition, chemotherapy, drug bioavailability enhancement, and radiotherapy with various drugs or factors, attracted our special interests. To prove the efficacies of SSD when treated in combination with docetaxel, a combination index assay, synergy score of matrix assay, and xenograft model were conducted.

In the Combination Index assay (Figure 6), the IC_50_ values of SSP and docetaxel in anti-MDA-MB-231 cells are 70.48 μg/mL, and 1.85 nanomolar, respectively. When treated simultaneously, the IC_50_ values of SSP and docetaxel decreased to 4.73 μg/mL, and 1.18 nanomolar, respectively. The Combination Index is 0.70, which means there is a synergism of SSP and docetaxel in anti-MDA-MB-231 cells. The consistent results can be gained by the Synergy score detection assay where the mean value of the synergy score calculated by the ZIP method is 5.79, with the most synergistic area score of 20.68. In the combination of luteolin and docetaxel, the mean value of the synergy score is 7.217, and the synergy score of the most synergistic area (White rectangle) is 19.58 (Figure 7).

In-vivo assay (Figure 8), there are no significant differences among the Vehicle control group, DTX group, and SSP-L group in terms of tumor volume. However, the tumor volume of the combination group of docetaxel treatment at low dose plus the SSP treatment at low dose was significantly less than that of the Vehicle control group, which means there is supper-additivity between SSP and docetaxel in anti-TNBC.

## 3. Discussion

For the first time, a workflow (Figure 1) for both toxicity and drug interaction prediction of herbal medicine based on virtual screening and text mining [45,46] was constructed. For studies on drug toxicity, drug-drug interactions, and drug-food interactions, with detailed related information retrieved, this workflow is beneficial for hypothesis construction and insight interpretation. Moreover, it has many superiorities over fixed prediction models. First, no prediction model can predict the toxicity and drug interactions like this workflow at the same time, which is important for a comprehensive safety assessment of complex drug mixture. Second, drug-drug interaction prediction models can only be utilized for drug pairs of interest, where the name of the interested drug pair should be offered in advance. However, this workflow can show some insights into drug combinations of >2 compounds without any pre-purpose needed. Even, the interaction between active compounds and some other factors such as ionizing radiation or carcinogens can be indicated by this workflow. Third, no detailed insights or interpretations of any toxicity prediction model are available. But this text mining-based procedure will include data resources, clinical trial details, and even experiment protocols for a result assessment in addition to the endpoint alert. Forth, as more and more information is documented in the chemical databases, a flexible workflow show much more potential and comprehensive assessment of the compound toxicity and interactions compared to a fixed prediction model which is only trained for specific toxicities prediction and utilized in limited scenarios. Fifth, this workflow is based on a combined SwissSimmlarity score, which has been proved to show better performance compared to fingerprint as the unique structure descriptor in activity prediction.

There is limited information about toxicity and drug interactions of natural active compounds (Table 3) documented in public databases. In PubChem, the most powerful database of chemical information regarding toxicity, only 23 out of 495 active compounds of 20 representative herbs can be retrieved. So, it is not feasible to make a safety assessment on herbal medicines by searching through public databases. Although there are many expert systems constructed for drug toxicity prediction based on QSAR or molecular docking, most QSAR-based systems employing models trained by FDA-approved drugs may show less confidence in the prediction of natural products. Plus, there are huge knowledge gaps between the molecular docking result and drug toxicity for the distinguished roles of one protein in systematic toxicity and the roles of compound-protein interactions in the activation of proteins are elusive. To solve this problem, this workflow tries to predict the properties of the unknown compounds by their similar bioactive compounds based on optimal QSAR on the SwissSimilarity platform.

There are no available models for toxicity or drug-drug interaction prediction for a mixture of more than two different compounds because no dataset can be gained for this kind of model training. All drug interaction prediction models are just trained for drug pairs of interest [47,48,49]. Although there is some interaction information of three-compound combinations retrieved in this study, even based on the most powerful database for chemical information-PubChem, most drug interaction retrieves are also documented for drug pairs. So, there is still a knowledge gap between the predictions of this workflow and the final clinical performance of plant complexes. All the predictions should be proved by experiments. The prediction result should be treated as preliminary hypotheses.

For a logistic regression model evaluation, metrics of sensitivity and specificity are the most introduced. Receiver operating characteristic curves (ROC), graphs of the specificity vs. the sensitivity, dependent on different thresholds, can show the performance of various models trained by the same dataset. Moreover, the area under the ROC curve can be used to compare different models trained by various algorithms and strategies. However, for the toxicity and drug interaction prediction of natural compounds, there are insufficient data for a logistic regression model training. Here we employed a combined system as beforementioned. To evaluate this system, two new concepts FEP-SS and FCP-SS (Defined as Box 1) were introduced to find the similarity range corresponding to consistent predictions or uncontroversial predictions compared to the retrieve with the highest similarity, respectively. More similar compounds and the toxicity and drug interaction information can be retrieved for active compounds with a relatively large similarity range because there is a significantly negative correlation between information retrieved and the similarity threshold (Figure 2a). The higher the similarity threshold is set, the more precise the prediction of this system is, but the less active compounds can be predicted (Figure 3b and Figure 4c,d). Ambiguous predictions are acceptable for the toxicity prediction of natural compounds, so the mean value of FCP-SS was set as the similarity score threshold. As more and more data are documented in the PubChem database, theoretically, a “stricter” similarity threshold for the prediction of most active compounds can be set in the future.

There are no distinct conclusions about the relationship between the content of an active compound and the weight of the compound on the activity of plant complexes. Some compounds accounting for a small proportion of the total herbal medicine may still exert remarkable activities, while some compounds accounting for a large proportion of total extracts may show little bioactivities. Given the drug interactions, situations will be much more complex, that is where the significance of this manuscript comes from. It filled a vacancy in conventional network pharmacology which lacks drug toxicity and interactions analysis in a complex system.

In QSAR analysis, chemical similarities such as fingerprints [50,51] and shape similarity [37] are the most popular descriptors for the structure of small molecules. In general, fingerprint similarity performs better than shape similarity in terms of bioactivity prediction [40]. That is why most toxicity prediction-expert systems take fingerprints of molecules as the structure descriptors. But shape similarity, independent of fingerprint similarity, can bring some extra information for indicating the chemical structure of drugs. So, a combined model trained by machine learning plays better performance in terms of structure-activity relationship prediction [40]. Herein, we tended to employ the combined model offered by the SwissSimilarity platform, a user-friendly platform with the “Bioactive”-compound class and several compound libraries for natural compound-data mining, to find the similar compounds with similar bioactivities [38]. For the stage of endpoint alert, most expert systems show limited information, which is difficult for prediction and insight interpretation. To solve this problem, ligand-based virtual screening outperforms an ambitious prediction model. However, this workflow was much more time-consuming for manual prediction interpretation compared to conventional expert systems. And it needs professional knowledge to interpret toxicity information to avoid an interpretation error. Text-classification and interpretation models trained via machine learning may solve this problem someday.

There are synergistic effects of crude extracts of SSD combined with docetaxel in anti-TNBC. Luteolin in concomitant use of docetaxel was also proved to show super-additive effects in anti-TNBC cells at certain doses (Figure 7). These experimental results are consistent with the prediction results. Till now, there are no effective medications but chemotherapeutic drugs for the management of triple-negative breast cancer [52,53]. However, most chemotherapy will incur more than 20 different severe adverse events such as anemia, diarrhea, fatigue, nausea, vomiting, and hair changes [4]. Based on the potential synergism of SSD and docetaxel, with less toxicity, lower therapeutic doses of combination treatment of docetaxel and SSD may accomplish the counterpart even better efficacies compared to the independent treatment of docetaxel or SSP.

## 4. Conclusions

For the first time, a workable and dependable workflow of in-silico drug toxicity and interaction prediction for plant complexes was built. From the PubChem database, 31 different sections of toxicity information such as “Acute Effects”, “NIOSH Toxicity Data”, “Interactions”, “Hepato-toxicity”, “Carcinogenicity”, “Symptoms”, and “Human Toxicity Values” sections have been retrieved, with dozens of active compounds predicted to exert potential toxicities. In *Spatholobus suberectus* Dunn (SSD), there are 9 out of 24 active compounds predicted to play synergistic effects on cancer management with various drugs or factors, which is consistent with the experimental data.

## 5. Materials and Methods

### 5.1. Dataset Assembly

A dataset containing the active compounds of 20 herbal medicines was gathered from the TCMSP database [54], based on the ADME criteria ((“Oral bioavailability” ≥ 0.3 and “Drug-likeness” ≥ 0.18), given all the herbal medicines are presumed to be administrated orally. Finally, the dataset “active_comp_pool_tcmsp.csv” contained 561 active compounds in total (495 distinct active compounds), 13 active compounds of Fructus ligustri Lucidi, and 24 active compounds of Spatholobus suberectus Dunn, 20 active compounds of Hedysarum multijugum Maxim, and 92 active compounds of Licorice, respectively.

### 5.2. Similar Compound Data Mining

From the PubChem database, the mining of properties of active compounds was conducted firstly through a script coded in Python 3 (version 3.8.10) called “compound_properties_mining.py” using pubchempy (version 1.0.4) and pandas (version 1.2.5) packages. This script iterates over the “active_comp_pool_tcmsp.csv” dataset, specifically, the “Molecule Name” column, while fetching one “Molecule Name” at a time. The gathered property data of active compounds were written to a CSV file named “active_comp_proper_pubchem.csv”. After duplicate values deletion, the mining of similar compounds of active compounds was done through the web scraper script called “similar_comp_crawler.py”. This script iterated the “Active_compound_name” column and the “isomeric_smiles” column of the dataset storing the properties of active compounds. The isomeric SMILES code is posted as a query to the SwissSimilarity website (updated version issued in Dec. 2021), selecting “Bioactive” compound class, choosing “ChEMBL (actives only)” natural product library [55], based on combined methods [40]. All the data of similar compounds were stored in the file named “similar_comp_pool_swiss.csv”.

### 5.3. Toxicity and Drug Interaction Information Mining

Before toxicity and drug interaction information mining, using a script called “similar_compound_properties_mining.py”, the properties of similar compounds were collected with a similar method as the mining of properties of active compounds beforementioned and were stored in the file named “similar_comp_properties_sum.csv”. Then the toxicity and drug interaction information mining was conducted through the web scraper script called “toxicity_mining_pubchem.py”. After redundant-value deletion, all the toxicity and drug interaction information were stored in the file named “Toxi_infor_sum.csv”. Screened from the “Toxi_infor_sum.csv” file, the drug interaction information retrieved was separated and split into one “interaction” retrieve per row using a script named “drug_interactions_split.py” for further manual interpretation. The split data was stored in the file named “drug_interaction_pred_0.6171.csv”.

### 5.4. Prediction Interpretation

The final prediction of the toxicity or drug interactions of active compounds of 4 representative herbal medicines was interpreted manually, based on the toxicity information and drug interaction data of its similar compounds, with a reasonable similarity score threshold. The definitions of “Active compound”, “Prediction yield”, “Drug interaction”, “First elusive prediction-similarity score”, and “First contrast prediction-similarity score” can be referred to Box 1. Every row in the dataset “Toxi_infor_sum.csv” was regarded as one retrieve. In a retrieve, for certain toxicity annotation, the similar compound was annotated as toxic if there was at least one in 32 sections clarifying the certain toxicity of similar compounds, or regarded as ambiguously toxic if there were controversial insights about the certain toxicity, or regarded as non-toxic or anti-toxic if all the available information indicating it was non-toxic or anti-toxic, or documented as “N.A.” if there were no related insights or evidence in all sections. The prediction results of active compounds were indicated by the indicators (Similar compounds with a similarity score above the similarity score threshold). For certain toxicity predictions, the active compound was regarded as toxic if major indicators were annotated toxic, or regarded as ambiguously toxic if there were controversial insights about indicators, or regarded as non-toxic or anti-toxic if all the indicators were non-toxic or anti-toxic, or documented as “N.A.” if there were no related insights or evidence about all the indicators of one active compound. The basic statistical analysis of this prediction study was done through the scripts named “parameter_similar_comp_properties.py”, and “parameter_comp_toxicity.py”. The predicted drug interaction network of active compounds in Spatholobus suberectus Dunn for cancer management was made by Cytoscape (Version 3.8.2) [56].

Box 1Definition of the basic concept in the prediction system.1.Active compound The active compounds of herbal medicine defined here, are the natural products documented in the TSCSP database for a certain herbal medicine, screened out based on the criteria (“Oral bioavailability” ≥ 0.3 and “Drug-likeness” ≥ 0.18).2.Drug interactions Drug interactions, in such a prediction system, include drug-food interactions, drug-drug interactions, and interactions of drugs with other factors such as carcinogens and ionizing radiation.3.Prediction yield In such a prediction system, the prediction yield is defined as the number of active compounds of herbal medicines with at least one kind of toxicity or drug interaction information predicted.4.First elusive prediction-similarity score (FEP-SS) Among all the information retrieved of similar compounds, for a certain toxicity prediction of an active compound based on such a system, as the similarity score decreases, the first elusive prediction-similarity score is the similarity score corresponding to the first elusive, arguable, or equivocal toxicity information retrieve compared to the toxicity information retrieve with the largest similarity score.5.First contrast prediction-similarity score (FCP-SS) Among all the information retrieves of similar compounds, for a certain toxicity prediction of an active compound based on such a system, as the similarity score decreases, the first contrast prediction-similarity score is the similarity score corresponding to the first contrast toxicity information retrieve compared to the toxicity information retrieve with the largest similarity score.

### 5.5. Preparation of Spatholobus Suberectus Dunn-Percolation (SSP) Extract

SSP was prepared and made a quality control as before described, its chemical profile can be referred to in previous studies [57]. Dried SSD stems were ground into coarse powder, then it was extracted using a percolating device with 10 times volumes (*v*/*w*) of 60% ethanol. The filtrate was then concentrated under reduced pressure by a rotary evaporator. The concentrated percolation extracts were then freeze-dried (Percent yield 20%) and stored at 4 ℃ for further use.

### 5.6. Cell Culture and Treatment

MDA-MB-231 cells were obtained from American Type Culture Collection (Manassas, VA, USA). All cells were maintained in glucose-containing (4.5 g/L) Dulbecco’s modified Eagle medium (Gibco, Grand Island, NY, USA), supplemented with fetal bovine serum (10% *v*/*v*, Gibco, Grand Island, NY, USA), penicillin (Sigma-Aldrich, St. Louis, MO, USA, 100 U/mL), and streptomycin (Sigma-Aldrich, St. Louis, MO, USA, 100 µg/mL) in a humidified atmosphere of 5% CO2 at 37℃. Cells were seeded onto 96-well plates at the density of 3–5 × 10^3^/well. After undergoing serum starvation for 24 h, they were treated with different concentrations of SSP, luteolin (DIECKMANN (HK) CHEMICAL INDUSTRY COMPANY LTD, Hong Kong, China), or docetaxel (Beijing Aosaikang Pharmaceutical Co., Ltd., Beijing, China). The tumor cell growth inhibitory effects of drugs were detected by CellTiter 96^®^ AQueous Non-Radioactive Cell Proliferation Assay containing 3-(4,5-dimethylthiazol-2-yl)-5-(3-carboxymethoxyphenyl)-2- (4-sulfophenyl)- 2H-tetrazolium) (MTS) kit (Promega, Wisconsin, DA, USA) as per the manufacturer’s protocol. The IC_50_ values of drugs were calculated by linear or nonlinear regression. The Combination Index was calculated after 40 h- of drug treatment using the formula [58]: Combination Index = (D)1/(Dx)1 + (D)2/(Dx)2, where (Dx)1, (Dx)2 are the concentrations of the tested substance 1 and the tested substance 2 used in the single treatment that was required to decrease the cell viability by x%, and (D)1, (D)2 are the concentrations of the tested substance 1 in combination with the concentration of the tested substance 2 that together decreased the cell viability by x%. The synergy score was calculated on the SynergyFinder platform (http://www.synergyfinder.org/) [59] with “Matrix” format and inhibition-Phenotypic Response, using the ZIP method [60] after 24 h-drug treatment.

### 5.7. Xenograft Model

The xenograft model was constructed as before described [57]. Female (BALB/c) nude mice (6–7 weeks old) were purchased from Harlan Laboratories, Indianapolis, IN, USA that were housed and maintained in the Laboratory Animal Unit, the University of Hong Kong, a specific pathogen-free and climate-controlled room (22 ± 2 °C, 50 ± 10% relative humidity) with a 12-h light/dark cycle and provided with diet and water ad libitum. MDA-MB-231 cells (2 × 10^6^/site) were implanted subcutaneously into the bilateral flank of each mouse. Palpable and measurable tumors were initially found 10 days after cell injection. Then, the animals were randomly assigned into five groups that were received the following treatments: the Vehicle control group (*n* = 6) received Milli-Q water; the SSP-L group (*n* = 6) received SSP (0.4 g/kg/p.o, daily); the SSP-H (*n* = 6) group received SSP (0.8 g/kg/p.o, daily); the DTX group (*n* = 6) received docetaxel (2.5 mg/kg/i.p. week); the combination group (DTX & SSP-L) (*n* = 6) received docetaxel (2.5 mg/kg/i.p. week) plus SSP (0.4 g/kg/p.o, daily). The tumor size was calculated using the formula: 0.5 × lengths × width2. All experiments were approved by the Institutional guidelines of Laboratory Animal Care and Committee on the Use of Live Animals in Teaching and Research (CULATR No.: 4484-17).

### 5.8. Statistical Analysis

Linear or non-linear regression was operated with GraphPad Prism 7 (GraphPad Software, San Diego, CA, USA) choosing log(inhibitor) vs. response-Variable slope (four parameters) as the equation. All data were expressed as Mean ± SD or Mean ± SEM. One-way ANOVA was employed to make a difference analysis for multiple groups’ comparation. The difference between two groups was analyzed by a two-tailed Student’s *t*-test. Significance was established at *p* < 0.05.

## Figures and Tables

**Figure 1 ijms-23-10056-f001:**
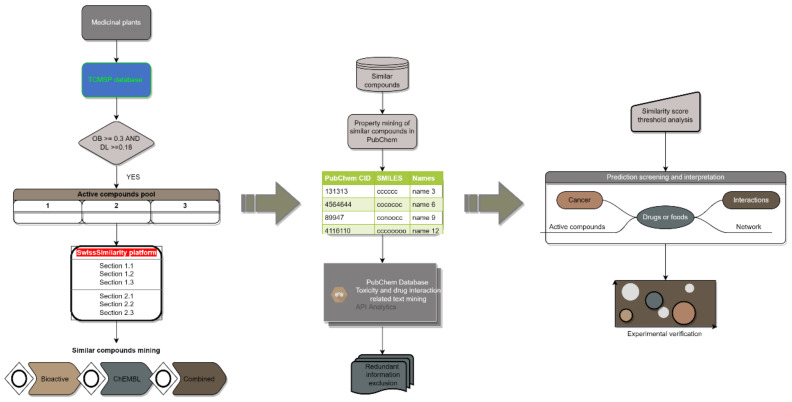
Workflow of in-silico toxicity and drug interaction analysis based on chemical similarity. Active compounds of herbal medicines are screened out based on some criteria. Then, as a query for mining similar compounds, the SMILES of active compounds will be collected. Based on the properties of similar compounds, the information on toxicity and drug interactions of similar compounds are retrieved. The final predictions and interpretations of active compounds will be made on a reasonable similarity score threshold. Experiments will be conducted to demonstrate the prediction results.

**Figure 2 ijms-23-10056-f002:**
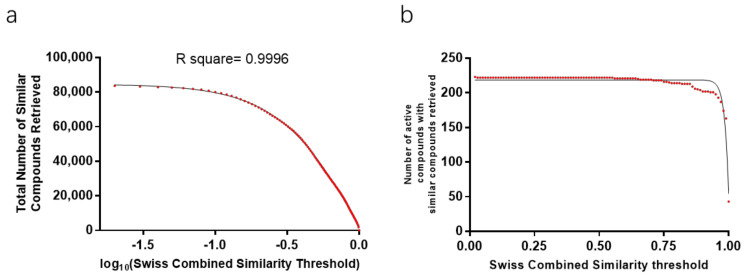
Similar compounds properties retrieved through computer programming for 20 herbal medicines. (**a**) Retrieve number curve of similar compounds properties dependent on similarity threshold. There is a significantly negative correlation between similar compounds retrieved and the similarity threshold. (**b**) Number curve of active compounds in herbal medicines for which the corresponding similar compounds were retrieved dependent on similarity threshold.

**Figure 3 ijms-23-10056-f003:**
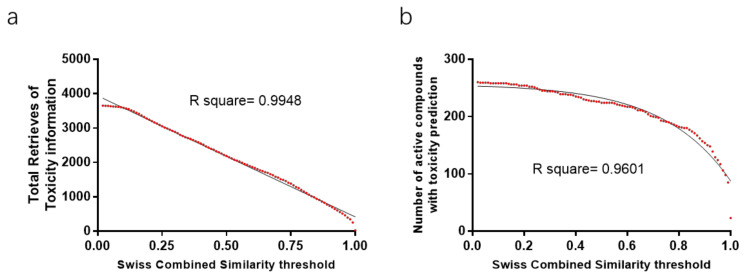
Toxicity information retrieved for 20 representative herbal medicines on different similarity thresholds. (**a**) Retrieve curve of toxicity information dependent on similarity threshold. There is a significantly negative correlation between the total toxicity information retrieved and the similarity threshold. (**b**) Number curve of active compounds in the 20 representative medicinal plants with toxicity information retrieved on different similarity thresholds. There is a negative correlation between the number of active compounds with toxicity information retrieved and the similarity threshold.

**Figure 4 ijms-23-10056-f004:**
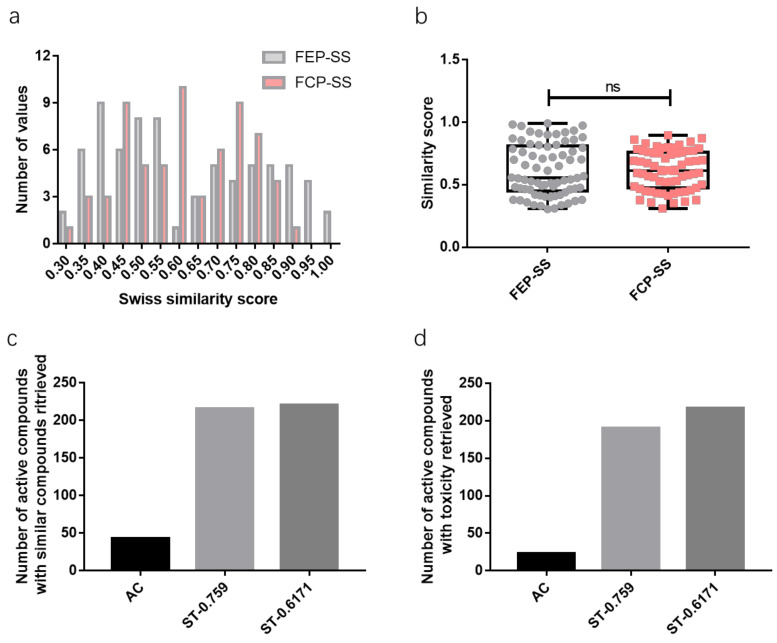
Statistical analysis of in-silico drug toxicity and interaction prediction for 20 different herbal medicines. (**a**) Histogram of FEP-SS and FCP-SS values of >0.3; (**b**) Box plots of FEP-SS and FCP-SS values of >0.3 for toxicity prediction; (**c**,**d**) The different amounts of active compounds with similar compounds or toxicity predictions based on different Swiss combined similarity score thresholds. AC: The group of toxicity information mining using active compounds of medicinal plants on PubChem platform, ST-0.759: Toxicity prediction group based on similarity score threshold of the third quartile value of FCP-SS (0.759), ST-0.6171: Toxicity prediction group based on a threshold of the mean value of P-SS (0.6171).

**Figure 5 ijms-23-10056-f005:**
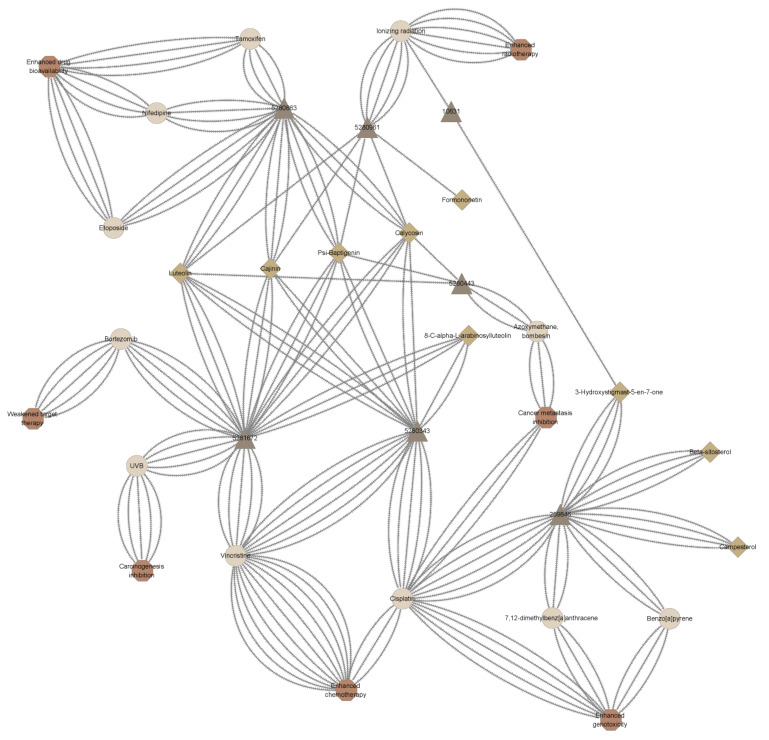
The network of cancer management-related interactions of active compounds in *Spatholobus suberectus* Dunn with drugs, ionizing radiation, or carcinogens, predicted through chemical similarity. Cajinin, calycosin, formononetin, luteolin, and psi-Baptigenin may potentiate the sensitivity of cancer cells to ionizing radiation. Luteolin enjoying similarity of 0.999 to apigenin was predicted to inhibit cancer metastasis. 8-C-alpha-L-arabinosylluteolin, and luteolin probably not only suppress UV-induced skin tumorigenesis but also have potential synergistic effects when treated in combination with many chemotherapeutic drugs. Cajinin, calycosin, luteolin, and psi-Baptigenin, similar to kaempferol, were predicted to enhance the chemotherapeutic drug bioavailability because of the inhibition effects on P-glycoprotein (P-gp) and cytochrome P450 (CYP). However, there is a theoretical risk of enhanced genotoxicity using cisplatin with luteolin or 8-C-alpha-L-arabinosylluteolin supplements. In addition, weakened target therapy of bortezomib may occur due to the combined treatment of 8-C-alpha-L-arabinosylluteolin or luteolin.

**Figure 6 ijms-23-10056-f006:**
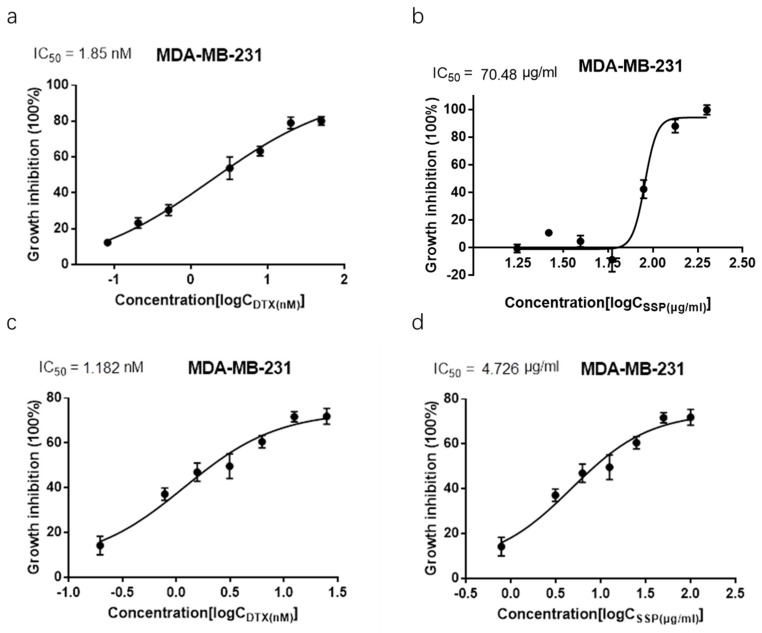
Combination index analysis for the synergistic effects of SSP and docetaxel. (**a**) MDA-MB-231 cells treated with different doses of docetaxel independently for 48 h, IC_50_ (DTX) = 1.85 nM, R square = 0.9805. (**b**) MDA-MB-231 cells treated with different doses of SSP independently for 48 h, IC_50_ (SSP) = 70.48 μg/mL, R square = 0.9854. (**c**,**d**) MDA-MB-231 cells treated with different doses of SSP (μg/mL)/docetaxel (nM) (100/25; 50/12.5; 25/6.25; 12.5/3.13; 6.25/1.56; 3.13/0.78; 0.78/0.20) for 48 h, IC_50_ (DTX) = 1.18 nM, IC_50_ (SSP) = 4.726 μg/mL, R square = 0.9424. CI ≈ 0.70, which means there is a synergism between SSP and docetaxel.

**Figure 7 ijms-23-10056-f007:**
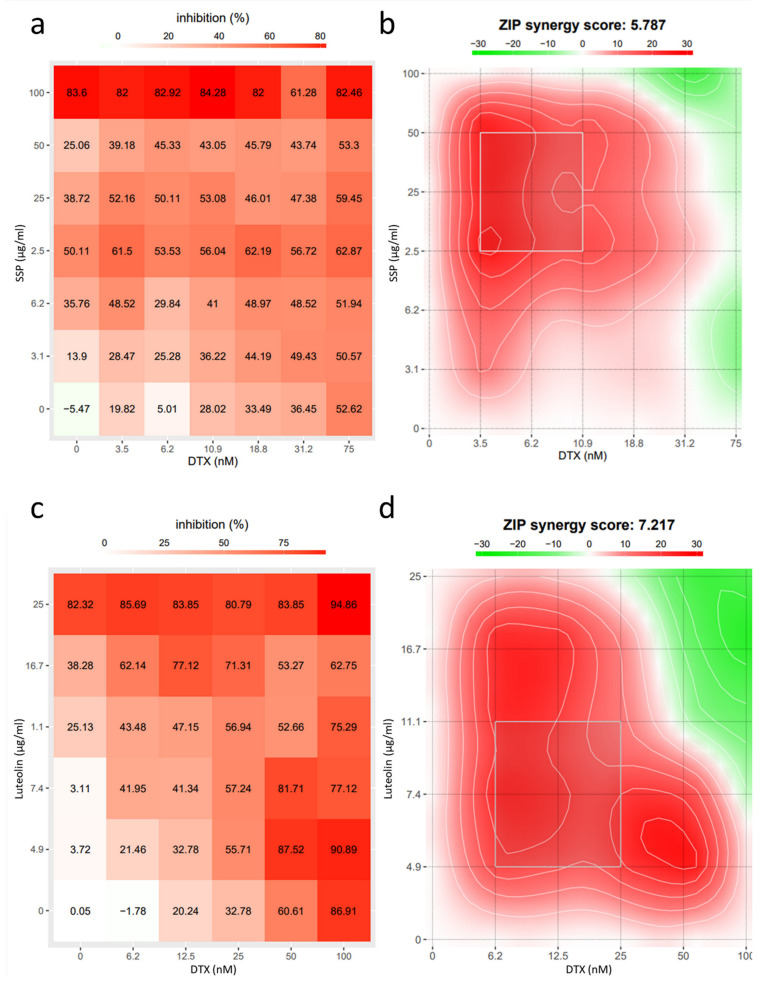
Quantitative analysis of the synergistic effect of SSP or luteolin combined with docetaxel through the ZIP method on the growth inhibition of MDA-MB-231 cells. (**a**,**c**) 2-D heat map of the dose-response matrix (Inhibition ratio) of drugs. (**b**,**d**) 2-D heat map of synergy score. In the combination of SSP and docetaxel, the mean value of the synergy score is 5.79, and the synergy score of the most synergistic area (White rectangle) is 20.68; In the combination of luteolin and docetaxel, the mean value of the synergy score is 7.217, the synergy score of the most synergistic area (White rectangle) is 19.58. The data were gained from two independent experiments.

**Figure 8 ijms-23-10056-f008:**
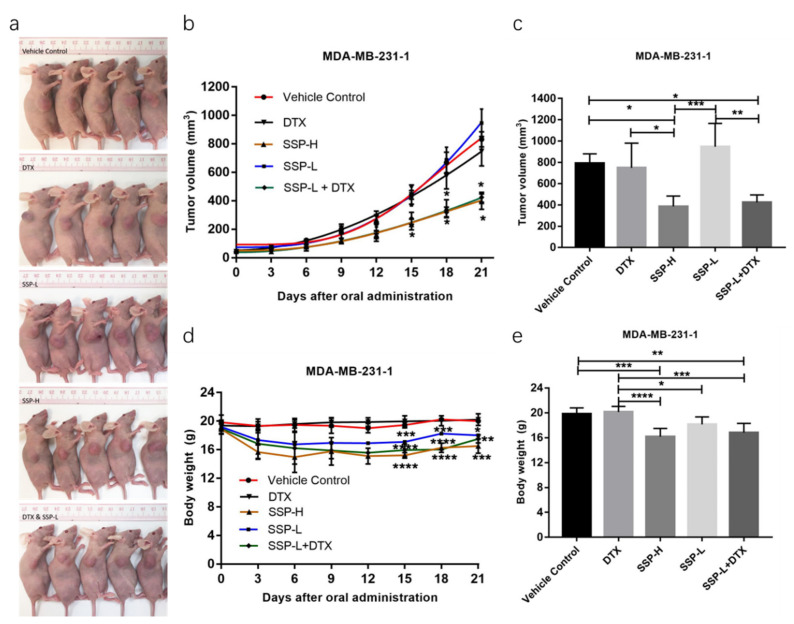
Qualitative analysis of the synergistic effects of SSP and docetaxel in anti-TNBC. (**a**) Representative pictures of mice xenograft with different treatments for 21 days. (**b**) Tumor volume curve. The Vehicle Control group received oral administration of Milli-Q water; The SSP-L group received oral administrations of SSP (0.4 g/kg/day); The SSP-H group received oral administrations of SSP (0.8 g/kg/day). The DTX group received administration of docetaxel (i.p., 2.5 mg/kg/week). The DTX & SSP-L group received oral administrations of SSP (0.4 g/kg/day) and docetaxel (i.p., 2.5 mg/kg/week). Data are shown as mean ± SEM (*n* = 6) with two independent experiments. (**c**) Statistical analysis of tumor volume at the endpoint of different mice with different treatments aforementioned. Data are shown as mean ± SEM (*n* = 6). * *p* < 0.05 (Vehicle control vs. SSP-H; Vehicle control vs. SSP-L + DTX; DTX vs. SSP-H), ** *p* < 0.01 (SSP-L vs. SSP-L + DTX) and *** *p* < 0.001 (SSP-L vs. SSP-H). (**d**) Bodyweight detection of the xenograft model experiment. Data are shown as mean ± SD (*n* = 6) with two independent experiments. (**e**) Statistical analysis of mice body weight at the endpoint of different groups. Data are shown as mean ± SD (*n* = 6). * *p* < 0.05 (DTX vs. SSP-L), ** *p* < 0.01 (Vehicle control vs. SSP-L + DTX), *** *p* < 0.001 (Vehicle control vs. SSP-H; DTX vs. SSP-L + DTX), **** *p* < 0.0001 (DTX vs. SSP-H).

**Table 1 ijms-23-10056-t001:** The toxicity information summary of similar compounds of representative active compounds.

Active Compound Name	Similarity Score	PubChem CID of Similar Compounds	Hepatotoxicity	Carcinogenicity
3-Hydroxystigmast-5-en-7-one	0.992	6010	0	N.A.
3-Hydroxystigmast-5-en-7-one	0.986	10631	0	N.A.
3-Hydroxystigmast-5-en-7-one	0.976	5997	N.A.	−1
3-Hydroxystigmast-5-en-7-one	0.917	6917715	0	N.A.
3-Hydroxystigmast-5-en-7-one	0.540	54454	0	N.A.
3-Hydroxystigmast-5-en-7-one	0.520	53232	0	N.A.
3-Hydroxystigmast-5-en-7-one	0.498	5280453	−1	N.A.
3-Hydroxystigmast-5-en-7-one	0.302	445354	1	N.A.
beta-sitosterol	0.999	5997	N.A.	−1
beta-sitosterol	0.804	5280453	−1	N.A.
beta-sitosterol	0.588	445354	1	N.A.
beta-sitosterol	0.588	445354	1	N.A.
campesterol	0.999	5997	N.A.	−1
campesterol	0.836	5280453	−1	N.A.
campesterol	0.594	445354	1	N.A.

1: Toxic; 0: Ambiguous; −1: Non-toxic or anti-toxic; N.A.: Not applicable.

**Table 2 ijms-23-10056-t002:** Similarity score corresponding to the first elusive or contrast prediction for certain toxicity of active compounds of four herbal medicines.

Active Compound Name	PubChem CID of Similar Compound	FEP-SS (Hepatotoxicity)	FCP-SS (Hepatotoxicity)	FEP-SS (Carcinogenicity)	FCP-SS (Carcinogenicity)
(+)-catechin	2369	0.472	-	-	-
(20S)-Dammar-24-ene-3beta,20-diol 3-acetate	5280453	0.31	-	-	-
18alpha-hydroxyglycyrrhetic acid	10133	0.331	-	-	-
3,22-Dihydroxy-11-oxo-delta(12)-oleanene-27-alpha-methoxycarbonyl-29-oic acid	5281004	0.265	-	-	-
3-Hydroxystigmast-5-en-7-one	5280453	0.498	-	-	-
DFV	4764	-		0.558	-
Glabranin	16078	0.703	-	-	-
Glabrene	3005573	-	-	-	0.353
Kanzonol F	441140	0.212	-	-	-
Medicarpin	441140	-	0.561	-	-
Olitoriside	54687/12560	0.337	0.449	-	-
Psi-Baptigenin	6237	-	0.176	-	-
Stigmasterol	445354	-	0.468	-	-
Aloe-emodin	42890/3059	0.463	0.413	-	-
Beta-sitosterol	445354	-	0.588	-	-
Campesterol	445354	-	0.594	-	-
Hederagenin	10133	0.486	-	-	-
Liquiritin	30323	-	-	0.379	-
Naringenin	5281576	-	-	0.477	-
Sitosterol	445354	-	0.588	-	-

FEP-SS: First elusive prediction-similarity score; FCP-SS: First contrast prediction-similarity score; -: Not applicable.

**Table 3 ijms-23-10056-t003:** Toxicity and drug interaction data mining in PubChem by active compounds of Fructus ligustri Lucidi, Spatholobus suberectus Dunn, Hedysarum multijugum Maxim, and Licorice.

PubChem CID	Active Compound Name	AE	Is	AET	HTE	NHTE	CC	PSR	HT	EC	NHTV	OTS	NTPS
5280448	Calycosin	-	+	+	+	+	-	-	-	-	-	-	-
9064	(+)-catechin	+	-	-	-	-	-	-	-	-	-	-	-
222284	Beta-sitosterol	+	-	-	-	-	-	-	-	-	-	-	-
73299	Hederagenin	+	-	-	-	-	-	-	-	-	-	-	-
5280863	Kaempferol	-	+	+	+	+	+	+	-	-	-	-	-
5280445	Luteolin	+	-	-	-	-	-	-	-	-	-	-	-
5280343	Quercetin	+	+	+	+	+	+	+	+	+	+	+	+
5280794	Stigmasterol	-	-	+	+	-	-	-	-	-	-	-	-
439533	Taxifolin	+	-	-	-	-	-	-	-	-	-	-	-

AE: Acute Effects; Is: Interactions; AET: Antidote and Emergency Treatment; HTE: Human Toxicity Excerpts; NHTE: Non-Human Toxicity Excerpts; CC: Carcinogen Classification; PSR: Populations at Special Risk; HT: Hepatotoxicity; EC: Evidence for Carcinogenicity; NHTV: Non-Human Toxicity Values; OTS: Ongoing Test Status; NTPS: National Toxicology Program Studies. +: with retrieves; -: no retrieves.

## Data Availability

All data generated or analyzed during this study can be available on GitHub (https://github.com/zhangfeng-2021/In-silico-drug-toxicity-and-interaction-prediction-for-plant-complexes) (accessed on 26 July 2022).

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
