# Peer review of "In-Silico Drug Toxicity and Interaction Prediction for Plant Complexes Based on Virtual Screening and Text Mining"

_ijms, 2022, doi:10.3390/ijms231710056_

Round 1

Reviewer 1 Report (Previous Reviewer 1)

The authors revised the manuscript according to the suggestions. I, therefore, recommend accepting it as it is.

Author Response

We appreciate your time, patience, and feedbacks that have helped us improve this manuscript significantly.

Reviewer 2 Report (New Reviewer)

The paper describes an in silico toxicity and drug interaction analysis based on chemical similarity of active compounds of herbal medicines. As highlighted by the authors, this is important to find potential synergistic or antagonist effects that can cause secondary effects on patients. The research line is well defined using several parameters and the main results are useful to a future translational scenario. Therefore, the present work is suitable for publication in the International Journal of Molecular Sciences.

I would like to suggest to the authors to review the English of the manuscript. Several sentences start with “but”/”and” which should be avoided.

Other minor points to be checked/corrected:

All the Figures (namely Figures 1, 4 and 5) should be referred in the text

Author Response

  1. Several sentences start with “but”/”and” which should be avoided.

Thanks for your comments and suggestions. Some modifications have been made according to your suggestions, which have been marked up using the “Track Changes” function. More revisions will be made if you have any further comments or suggestions.

  1. All the Figures (namely Figures 1, 4 and 5) should be referred in the text.

Thanks for your comments. Some modifications have been made according to your comments, which have been marked up using the “Track Changes” function. More revisions will be made if you have any further comments or suggestions.

This manuscript is a resubmission of an earlier submission. The following is a list of the peer review reports and author responses from that submission.

Round 1

Reviewer 1 Report

The manuscript by Zhang et al. reports a predictive analysis of drug toxicity and interactions between natural substances and chemotherapy medications. The authors had set a clear goal of investigating the synergistic effect of different compounds on cancer management and presented very interesting results on the synergism between Spatholobus suberectus Dunn (SSD), luteolin and docetaxel on triple-negative breast cancer. I think that this work is solid, especially with respect to the design of workflow for the in-silico analysis and experiments to support the predicted results regarding the combinatorial treatment of cancer. Therefore, I recommend this work for publication as is. My suggestions are as follows:

1. Please identify the sample size for the cell experiments in Figure 7.

2. In Figure 8, state clearly which groups were used for statistical analyses.

3. I would suggest having a separate conclusion section at the end of the manuscript.

Reviewer 2 Report

In this manuscript, the authors performed chemical similarity analysis, in vitro, and in vivo experiments related to a botanical extract. First they identified phytochemicals from a database, then searched for similar chemicals with toxicity data. They inferred that their phytochemicals will have similar toxicity based upon chemical similarity. The authors lastly performed some in vitro and in vivo experiments involving the botanical extract.

The figures in the manuscript are easy to read. However, the manuscript is very dense. There is a major disconnection between the chemical similarity analysis, in vitro, and in vivo studies. There is a lot of data and I think basic, key information is buried among lots of details. The manuscript needs to be significantly simplified and presented in a way to clearly link the various studies with rationale. My expertise matches the manuscript very well (spanning in vitro and in silico toxicology, phytochemicals, and chemical similarity), but I had major challenges understanding this manuscript and it was very disjointed.

The authors used a botanical extract (Spatholobus suberectus Dunn-percolation (SSP)) but it was not characterized in any way for phytochemical constituents. For this study to make any sense, the phytochemical constituents need to be identified and/or quantified. This is one of the most fundamental pieces of information for any study involving botanical extracts.

If I understand correctly, the authors identified chemicals similar to phytochemicals of interest that had toxicity data, and then inferred that the phytochemicals will have similar. This is a very broad assumption. There are many programs out there that specifically predict different forms of toxicity and they are more specific than simply looking at chemical similarity. Since “toxicity” is very broad it is hard to approach the analysis in this way.

Continuing from the previous comment: A fundamental concept in the manuscript is FCP-SS and FEP-SS. On page 4, it says “The FCP-SS of one active compound is the threshold for retrieves without any controversial results, while the FEP-SS is the threshold for that all the retrieves are consistent.” Similar definitions were provided in Box 1. If I understand correctly, the authors changed the similarity threshold until one of the toxicity values was a false negative? This is very hard to understand and I don’t know how you can even apply this approach to so many endpoints simultaneously—it doesn’t make any sense.

The initial data mining involved carcinogenicity, hepatotoxicity, genotoxicity, and developmental/reproductive toxicity data. How is this related to in vitro study (growth inhibition assays) or the in vivo study (tumor size)? There is a major disconnection here.

In the very beginning, the authors extracted compounds from TCMSP database using criteria for OB and DL. It sounds like these are related to ADME but these are not common abbreviations and I have no idea what they mean.